# The Regulation of Homologous Recombination by Helicases

**DOI:** 10.3390/genes11050498

**Published:** 2020-05-01

**Authors:** Eric Huselid, Samuel F. Bunting

**Affiliations:** Department of Molecular Biology and Biochemistry, Rutgers, The State University of New Jersey, Center for Advanced Biochemistry and Medicine, 679 Hoes Lane West, Piscataway, NJ 08854, USA; huselid.eric@gmail.com

**Keywords:** helicase, DNA repair, recombination, replication, transcription, anti-recombinase

## Abstract

Homologous recombination is essential for DNA repair, replication and the exchange of genetic material between parental chromosomes during meiosis. The stages of recombination involve complex reorganization of DNA structures, and the successful completion of these steps is dependent on the activities of multiple helicase enzymes. Helicases of many different families coordinate the processing of broken DNA ends, and the subsequent formation and disassembly of the recombination intermediates that are necessary for template-based DNA repair. Loss of recombination-associated helicase activities can therefore lead to genomic instability, cell death and increased risk of tumor formation. The efficiency of recombination is also influenced by the ‘anti-recombinase’ effect of certain helicases, which can direct DNA breaks toward repair by other pathways. Other helicases regulate the crossover versus non-crossover outcomes of repair. The use of recombination is increased when replication forks and the transcription machinery collide, or encounter lesions in the DNA template. Successful completion of recombination in these situations is also regulated by helicases, allowing normal cell growth, and the maintenance of genomic integrity.

## 1. Introduction

Homologous recombination (HR) is an essential cellular process, which is required to repair DNA double strand breaks (DSBs), restart collapsed replication forks, and for the rearrangement of genetic information from parental chromosomes during meiosis [1]. Helicases are the enzymes that bind to nucleic acids, and translocate along the molecule, separating base-paired regions using energy from ATP. Helicases have been shown to be necessary to carry out various steps of recombination, and also for regulation of the rate and outcomes of recombination (Table 1). Several of these proteins are classical helicases, which act to unwind single-stranded DNA, whereas others can be classed as translocases, which use their motor activity to move along double-stranded DNA regions [2]. In eukaryotes, recombination proceeds by resection of DNA ends to form a single-stranded region, which is initially bound by Replication Protein A (RPA), and subsequently by RAD51 [1,3]. The nucleoprotein filament loaded with RAD51 invades the homologous DNA regions, creating a displacement loop (D-loop), where sequences from the broken DNA molecule and homologous template pair as ‘heteroduplex’ DNA. A DNA polymerase can subsequently add nucleotides at the free 3′ end, to restore the sequence around the break site. Several competing pathways then complete the HR process by disassembly of recombination intermediates. RAD51-mediated recombination is also essential for DNA replication during S phase, by restoring stalled replication forks, and by restarting replication after fork collapse [4,5,6].

The importance of helicase activities for recombinational repair is demonstrated by the pathology that arises in individuals inheriting mutant copies of helicase genes. Homozygous mutations affecting the RECQ helicases, BLM and WRN, as seen in Bloom Syndrome and Werner Syndrome, respectively, cause defective DNA repair, tumor susceptibility and abnormal development [7,8,9,10]. Understanding the contribution of helicases to recombination is therefore essential to identify how these proteins enable normal growth of the cell.

## 2. Regulation of DNA Double-Strand Break Resection by Helicases

Repair of a DNA double-strand break by HR requires ‘resection’ of the broken DNA end, which involves the production of a single-stranded region at the 3′ end [1]. Resection is considered a key step that commits repair of a DSB to the HR pathway for repair. Both the regulation and mechanism of DSB resection involve the activity of specific helicases (Figure 1). The first steps of end resection involve the MRN complex (MRX in yeast), a heterotrimeric complex of Mre11, Rad50 and Nbs1, which has affinity for broken DNA ends. Association of the CtIP protein with MRN stimulates the nuclease activity of Mre11 to begin short-range resection, and create a small region of single-stranded DNA (ssDNA) at the break site. Certain DNA structures, such as G-quadruplexes, can inhibit the process of resection. Recruitment of the Pif1 helicase aids in unwinding these complex DNA structures, and improves the efficiency of HR [11]. The helicase RECQL4 impacts the process of resection in a cell cycle-dependent manner, by helping the MRN complex recruit CtIP during S and G2 phases of the cell cycle [12]. RECQL4 also suppresses use of HR during the G1 phase of the cell cycle, by favoring the competing NHEJ pathway [13]. In this case, RECQL4 binds to Ku70/Ku80, stabilizing them at the break site, and directing repair toward NHEJ. The RECQ helicase, WRN (Werner syndrome helicase), is also able to promote NHEJ through an interaction with Ku70. In addition to its helicase function, WRN also has exonuclease activity, which can remove modified nucleotides at DSBs to promote NHEJ [14,15,16,17,18]. The helicase SMARCAL1/HARP also promotes NHEJ at this step, by re-annealing DSB ends that have unwound, thereby facilitating the binding of Ku70 [19]. 

Once short-range end resection is achieved, the single-stranded overhang is lengthened through ‘long-range resection’, to create a more substantial ssDNA substrate for nucleoprotein filament formation and strand invasion [20]. Long-range resection is achieved by two parallel pathways. One of these pathways is dependent on the exonuclease, Exo1, which removes one strand of the DNA duplex in a processive manner in the 5′-3′ direction. The other pathway involves a RECQ helicase working in combination with DNA2. In budding yeast, Sgs1 is the key RECQ helicase for long-range resection, and, in vertebrates, Bloom Syndrome helicase (BLM) provides the equivalent activity [21,22,23,24,25]. This evolutionarily conserved approach to DSB resection involves the unwinding of the DNA duplex by the RECQ helicase, followed by nicking of the resected strand by an endonuclease activity in DNA2. Studies in vitro and in budding yeast have demonstrated that RPA (Replication Protein A) plays an important function in ensuring productive long-range resection [24,26]. The RECQ helicases, BLM and RECQL1, also appear to contribute to resection by stimulating the exonuclease activity of Exo1 at DSBs [24,27,28]. The ability of WRN to substitute for BLM for DNA2-dependent resection has been a matter of some debate. Initial studies indicated that WRN is not able to provide the necessary unwinding activity for resection [23]. Other reports have supported the idea that WRN is also able to mediate DNA2-dependent resection, potentially after activation by phosphorylation by the cell cycle-dependent kinase, CDK1 [29,30,31,32]. In addition to acting as a nuclease, DNA2 also has helicase activity. This helicase activity contributes to resection by removing single-stranded DNA products that are produced by the unwinding and cutting of DSBs by DNA2 with its RECQ helicase partner [33,34]

Several other helicases contribute to regulation of DSB resection. DNA Helicase B (HELB) counteracts unwinding of DSBs by RECQ helicases, using its 5′-3′ ssDNA translocase activity [35]. This repressive effect on resection is exerted during G1 phase of the cell cycle. During the transition to the S/G2 phase of the cell cycle, HELB is exported from the nucleus, allowing increased resection and use of the HR pathway. The yeast SNF2-type ATP-ase, Fun30, and its mammalian ortholog, SMARCAD1, contribute to extensive resection of DSBs [36]. These proteins use ATP-dependent translocase activity to move along chromatin in the vicinity of DSBs, remodeling chromatin to allow extensive resection and normal HR [37,38]. Loss of either Fun30 or SMARCAD1 therefore makes cells hypersensitive to DSBs induced by treatment with the topoisomerase inhibitor, camptothecin. FANCJ is found alongside BLM during long-range resection, and appears to contribute to the efficiency of resection by stabilizing BLM, or by removing obstacles such as complex DNA structures [39,40]. FANCJ also helps recruit CtIP, thereby enabling the initial steps of resection [41]. The MCM8-9 helicase complex is also reported to promote resection [42], although other reports suggest it acts at a later stage of HR [43,44,45].

## 3. Single-Stranded DNA Binding Protein Displacement

3′ DNA overhangs produced through DSB resection become rapidly coated with single-stranded DNA binding proteins. In prokaryotes, SSB (single-strand binding protein) fulfills this role [1]. In eukaryotes, RPA, which is a heterotrimer of RPA70, RPA32 and RPA14, binds to single-stranded DNA overhangs. For recombination to take place, RPA must be replaced by RAD51, which is equivalent to RecA in prokaryotes. The nucleoprotein filament of single-stranded DNA coated with RAD51 forms a ‘presynaptic filament’, which pairs with homologous DNA regions through strand invasion, allowing formation of a displacement loop (D-loop) and template-based repair. In vertebrate cells, RAD51 paralogs (RAD51B, RAD51C, RAD51D, XRCC2, XRCC3 and DMC1) are also present in recombinogenic nucleoprotein filaments, and ensure efficient and productive recombination [46]. Assembly of the RAD51-containing nucleoprotein filament is therefore an essential step in recombination, and a step at which the rate of recombination can be regulated. Several helicase enzymes regulate the loading and removal of single-strand binding proteins (Figure 2). Prokaryotic UvrD displaces RecA, thereby limiting the rate of HR, and attenuating DNA damage signaling [47]. In budding yeast, the Srs2 helicase likewise acts as an ‘anti-recombinase’, by removing RAD51 from the nucleoprotein filament at resected DSBs [48,49]. Srs2 is also essential for regulating recombination during replication, and is recruited to SUMO-modified PCNA under conditions of replication stress [50,51]. By modulating the rate of HR, Srs2 may help prevent the appearance of defective HR intermediates [52].

Several helicases in vertebrates have been reported to act similarly to UvrD and Srs2, by reducing the rate of HR through displacement of single-stranded DNA binding proteins. A bioinformatic approach identified PCNA-Associated Recombination Inhibitor (PARI), a protein containing a UvrD-type helicase domain, which has a similar domain organization to Srs2, as a potential mammalian anti-recombinase [53]. Loss of PARI is reported to increase RAD51 loading at break sites, causing an elevated level of HR that is associated with genomic instability. PARI appears to stimulate the ATPase activity of RAD51, promoting RAD51 to dissociate from the DNA. The ability of PARI to displace RAD51 is quite weak, however, and only achieved with stoichiometric amounts of protein. The importance of the PARI helicase domain is likewise unclear, because it does not appear to have a functional ATPase activity. The exact mechanism for PARI-mediated regulation of HR is therefore still not fully characterized, and it may play a more important role in regulating other aspects of HR (See Section 4, below). In contrast, the Superfamily 2 helicase, FANCJ (Fanconi Anemia Complementation group J), is able to dissociate DNA complexes in a manner that is clearly dependent on ATP hydrolysis [54]. FANCJ displaces RAD51 from DNA in vitro, thereby reducing the efficiency of RAD51-dependent DNA strand-exchange. F-box DNA helicase 1 (FBH1) was originally shown to repress recombination in *S. pombe* [55,56]. Anti-recombinase activity of FBH1 was subsequently reported in vertebrates as well, including in human cells [57,58,59]. As with FANCJ, the ability of FBH1 to displace RAD51 is dependent on its helicase activity, which allows the FBH1 protein to move along the nucleoprotein filament, facilitating removal of RAD51. FBH1 also participates in an SCF complex, which ubiquitinates RAD51, leading to RAD51 removal [60].

RECQ helicases, such as Sgs1 and BLM, promote HR by mediating the ‘long-range’ resection of DSBs (see previous section). Several RECQ helicases also have anti-recombinogenic activity, however, which is mediated in part by removal of RAD51 from the presynaptic filament. Single-molecule imaging studies show that Sgs1 from budding yeast can displace RAD51 from DNA [61]. This function is conserved in mammalian species, as in vitro studies have demonstrated that the BLM helicase can also remove RAD51 from DNA [62]. Work in our lab showed that this anti-recombinase activity of BLM plays an important role in regulating the efficiency of HR in cells [63]. Cells that lack the HR factors, BRCA1 or BRCA2, normally show a substantial defect in RAD51 loading at DNA break sites, which results in defective HR and genomic instability. Co-deletion of BLM rescues this HR defect, by allowing increased accumulation of RAD51 at resected DSBs. In addition to BLM, RECQL5, another RECQ helicase, disrupts RAD51 filaments [64,65]. The loss of RECQL5 leads to hyper-recombination, which correlates with genomic instability and cancer susceptibility in *Recql5*^–/–^ mice.

The FIGNL1 (Fidgetin-like 1) helicase is required for normal HR, and this regulatory function appears to depend on the ability of FIGNL1 to displace RAD51 [66,67]. FIGNL1 is an AAA+ ATPase helicase, but the ATPase activity does not seem to be required for RAD51 displacement. In recent years, Polθ (DNA Polymerase theta) has also emerged as an important regulator of DNA repair [68]. Polθ has multiple enzymatic activities, including helicase activity. The helicase activity of Polθ is important for directing repair of DSBs toward ‘alternative end-joining’ instead of toward HR. This effect is achieved by removal of RPA from single-stranded regions of DNA by Polθ [69,70]. When Polθ is absent, rates of HR increase, and RAD51 accumulates at increased levels at break sites.

## 4. Dissolution of D-Loops

After the formation of a RAD51-loaded presynaptic filament, the broken DNA end pairs with a homologous DNA region, and begins a process called strand invasion [1]. The structure formed when an ssDNA filament invades a homologous DNA duplex is referred to as a displacement loop, or D-Loop (Figure 3). The formation of heteroduplex DNA, with the broken DNA molecule paired to the homologous template DNA, allows sequence at the break to be restored by a DNA polymerase enzyme. D-loops can be dissolved and displaced at various points by a number of different helicases which help to prevent erroneous strand invasion or limit the extent of polymerase activity. Yeast Srs2, in addition to displacing RAD51 from resected DNA ends [48,49], can dismantle D-loops, which helps to eliminate nonproductive recombination intermediates [71,72]. The disassembly of D-loops also prevents long-range DNA extension, reducing the frequency of formation of double-Holliday junctions intermediates, which can be resolved as crossovers. The Superfamily 2 (SF2) helicase Mph1 (Mutator Phenotype 1) acts in parallel with Srs2 to promote non-crossover outcomes to recombination [73]. The ability of Mph1 to dissociate D loops is strictly dependent on its helicase activity.

The homologs of Srs2 and Mph1 in mammals have similar D-loop dissociation activity. RTEL1 (Regulator of Telomere Length) is a putative Srs2 homolog that limits recombination by dissociating the D-loop recombination intermediates [74]. This effect of RTEL1 may be particularly important for suppressing recombination at telomeres, because RTEL1-knockout mouse cells show chromosome instability and telomere loss that is associated with embryonic lethality [75]. PARI, like Srs2, can promote non-crossover recombination by unwinding D-loops, which inhibits extension of the synaptic filament by Polymerase δ [76]. This activity of PARI appears to be independent of its UvrD helicase domain, therefore it is not known whether PARI works alone, or potentially in complex with some other factor. Fanconi anemia, a genetic disease that is associated with the loss of leukocytes and cancer predisposition, is caused by mutations in the FANC gene family. One of these genes, FANCM, is homologous to yeast Mph1, although FANCM has nuclease activity in addition to acting as a DNA helicase [77,78]. The nuclease activity of FANCM contributes to its role in repairing DNA damage caused by inter-strand crosslinking agents, but point mutants affecting the helicase domain do not seem to affect this repair pathway [79]. The helicase activity of FANCM instead appears to suppress crossover recombination, thereby preventing the exchange of genetic material between homologous DNA sequences during HR.

Of the RecQ helicases, yeast Sgs1 and mammalian BLM have been shown to have the ability to dissolve D-loops. Sgs1 acts as part of the Sgs1-Top3-Rmi1 complex to negatively regulate D-loops [80], and BLM forms an equivalent complex with TOPOIIIa-RMI1-RMI2 [81]. BLM therefore appears to counteract recombination at two potential steps, by displacing RAD51 from resected DNA ends as discussed in the previous section, and by dissociating D-loops. It is not clear which of these activities is most important for the contribution of BLM to maintenance of genomic integrity, and preventing the tumor susceptibility that is characteristic of Bloom Syndrome. WRN, the helicase that is mutated in Werner’s Syndrome, is required for genomic integrity and normal HR [82]. Genomic instability in WRN-deficient cells can be partially suppressed by the overexpression of a dominant-negative form of RAD51, suggesting that WRN normally resolves toxic recombination intermediates [83]. Biochemical evidence indicates that WRN degrades D-loops through branch migration dependent on its helicase domain, and by targeting 3′ strand-invaded DNA ends with its exonuclease activity [84,85,86]. RECQL1 can likewise dissociate D-loops using its branch migration activity [87]. At least three mammalian RECQ helicases therefore counteract formation of D-loops, although the relative importance of these activities for recombination in cells is not fully established.

## 5. Activities of Helicases in the Postsynaptic Stages of HR

After the formation of a stable D-loop, the synaptic filament is paired as heteroduplex DNA with a homologous region from another chromosome or chromatid. The next stages determine the outcome of recombination, which can proceed through either a ‘synthesis dependent strand annealing’ (SDSA) pathway or through the ‘double-strand break repair’ pathway (DSBR) [1]. SDSA involves unwinding the D-loop intermediate formed after strand invasion, and always produces non-crossover products. Helicases that can disrupt D-loops therefore tend to promote SDSA as a mechanism for completing HR. The DSBR pathway involves extended synthesis of DNA within the D-loop, second-end capture, and formation of ‘double-Holliday junction’ structures (Figure 4). The resolution of double-Holliday junctions produces a mixture of crossover products, in which regions from the broken DNA molecule and template duplex become spliced together, in addition to non-crossover products. The frequency with which these pathways are used varies in different cell types. A mixture of crossover and non-crossover recombination products are produced during the repair of Spo11-mediated DNA breaks in meiosis, but non-crossover outcomes are favored in somatic cells [88,89].

The MCM8-9 complex is proposed to act at the D-loop to unwind the template DNA, and allow the paired heteroduplex DNA to be extended by a polymerase [90]. MCM8 and MCM9 form a helicase complex that is paralagous to the MCM2-7 complex, which forms the primary eukaryotic replicative helicase [91]. MCM8-9 can mediate normal S phase DNA replication when MCM2-7 is absent [92], but is principally involved in HR-associated DNA replication [43,44,45]. Mouse knockouts of MCM8 or MCM9 are sterile, demonstrating the essential role of these factors in meiotic recombination during gametogenesis. The loss of MCM8 and MCM9 also disrupts HR in somatic cells, leading to hypersensitivity to DNA damage, genomic instability, and tumor predisposition [93,94]. MCM8-9 may act in a parallel pathway with the HELQ helicase to mediate HR after strand invasion, because cells lacking both HELQ and the MCM8-9 loading factor, HROB, show a very severe defect in HR and persistent RAD51 loading [90]. HELQ is a 3′-5′ helicase, which is required for normal DNA repair and for the survival of germ cell progenitors [95,96,97]. Work in *C. elegans* and mice has supported a model in which HELQ acts after RAD51 loading to enable HR [98,99]. In the absence of HELQ, RAD51 loads at recombination sites, but remains bound there, indicating that HELQ may be required for disassembly of RAD51 from the postsynaptic filament to complete recombination. The importance of HELQ for normal DNA repair is demonstrated by the high tumor incidence seen in HELQ^–/–^ mice.

RAD54 and ATRX are two members of the SWI/SNF2 class of SF2 helicases, which have important functions in regulation of events at several stages of recombination including those taking place after strand invasion [100,101]. RAD54 stimulates strand invasion of RAD51-loaded DNA into a homologous dsDNA duplex [102], and stabilizes RAD51 nucleoprotein filaments, but it can also displace RAD51 from DNA, dependent on its ATPase activity [103,104]. RAD54 mediates branch migration of Holliday junctions, and promotes resolution of Holliday junctions through stimulation of the Mus81 nuclease [105,106]. ATRX (alpha thalassemia/mental retardation syndrome X-linked) has an N-terminal chromatin-binding domain, and a C-terminal helicase domain [100]. Mutations in either of these domains cause a developmental disorder characterized by growth defects and intellectual disabilities, although mutations in the helicase domain tend to have a milder effect [107]. The chromatin-binding domain of ATRX helps load the variant histone H3.3 at repair sites, and the loss of this activity appears to contribute to DNA repair defects in ATRX-deficient cells. Loss of ATRX causes persistence of DNA breaks, and abnormal HR, as measured by cellular reporter assays [108]. RAD51 loads normally in ATRX-deficient cells, but a greater amount of DNA synthesis takes place after strand invasion, and crossover products from HR are suppressed. ATRX therefore appears to increase the use of the DSBR pathway for HR, instead of the SDSA pathway [3]. RAD54 and ATRX do not act as typical dsDNA-unwinding helicases. Instead, their helicase domains provide ATP-dependent translocase activity, allowing the proteins to move along DNA to regulate HR. The *S. cerevisiae* Mer3 helicase also promotes crossover formation in meiosis through the DSBR pathway, by stabilizing D-loops to promote extension of heteroduplex DNA in the 3′–5′ direction [109]. The mouse homolog of Mer3, HFM1, is required for crossover recombination during meiosis [110]. HFM1-knockout mice therefore show a failure to complete spermatogenesis and are infertile.

In yeast and mammals, RECQ helicases play vital roles at the late stages of HR by mediating branch migration and subsequent ‘dissolution’ of Holliday junctions. Holliday junctions can be ‘resolved’ by structure-specific nucleases, which make nicks around the Holliday junction, leading to formation of a mixture of crossover and non-crossover products [111]. In contrast, dissolution of Holliday junctions produces only non-crossover products, therefore RECQ helicases and their associated factors are very important regulators of the outcome of recombination. Branch migration refers to directed unwinding and re-annealing of DNA regions around the Holliday junction, to move the Holliday junction or alter the extent of heteroduplex DNA. This activity is carried out by Sgs1 in yeast, and BLM in mammalian cells [112,113]. Branch migration of double-Holliday junction intermediates can bring the two Holliday junctions together, to form a ‘hemicatenane’ structure that can be dismantled by Sgs1 or BLM in complex with several other proteins [80,114,115,116,117]. The protein complex for the dissolution of Holliday junctions has been called the ‘dissolvasome’, and is made up of Sgs1-Top3-Rmi1 in yeast, and BLM-TOPIIIa-RMI1-RMI2 in mammals. In particular, the topoisomerase activity provided by Top3/TOPIII allows nicking and re-ligation of the heteroduplex DNA in the hemicatenane to restore intact, separate DNA molecules. Through this process, Sgs1 and BLM greatly reduce the frequency of crossover recombination. Cells lacking BLM, such as those from Bloom Syndrome patients, show a much higher rate of crossovers, which can be demonstrated experimentally by the sister chromatid exchange assay, which allows exchanges of chromosome regions to be quantified by differential staining [118]. In yeast, proteins of the SIC/ZMM family promote crossovers during meiosis, by counteracting the ability of Sgs1 to dissolve Holliday junctions [119]. This indicates that regulation of helicases and helicase-containing complexes is essential to ensure appropriate outcomes of recombination.

In addition to BLM and Sgs1, other helicases are active in the final stages of HR. Deletion of either RECQL1 or RECQL5 is reported to increase the frequency of sister chromatid exchanges in BLM-knockout cells, suggesting that these alternative RECQ helicases act as a backup for BLM in suppressing crossover recombination [120]. As discussed previously, RAD54, RECQL1 and WRN have branch migration activity [84,87,105]. Branch migration activity has also been reported for SMARCAL1, ZRANB3 and FANCM [121,122,123,124], although these helicases may have specialized roles at replication forks instead of in general HR-mediated repair of DNA double-strand breaks.

## 6. Helicase-Mediated Regulation of Recombination at Sites of Replication

A recombination-based process called ‘break-induced recombination’ can restart replication forks that collapse at obstacles such as a single-strand breaks or DNA cross-links [125]. Replication forks do not immediately collapse on encountering obstacles, however, and can instead pause for a period of several hours, allowing a chance for any obstacles to be removed and for replication to continue [6]. Helicase activity contributes to stability of stalled replication forks, thereby reducing replication fork collapse and break-induced recombination (Figure 5). Electron microscopy has shown that stalled replication forks frequently become reversed to assume a ‘chicken foot structure’, which becomes loaded with RAD51 to promote stabilization. SMARCAL1 acts as an annealing helicase to bring single-stranded DNA molecules at the replication fork together, to promote reversal of stalled replication forks, and help protect them from endonucleolytic degradation [121,122,126]. The ZRANB3 helicase can also reverse stalled replication forks, and acts alongside SMARCAL1 to protect forks in the presence of oncogene-induced replication stress [127]. Another annealing helicase, TWINKLE, is required for replication of mitochondrial DNA [128]. TWINKLE activity is proposed to contribute to a form of recombination-mediated replication initiation, similar to replication by bacteriophage T4, which has been observed in mitochondria [129]. The exact mechanism of helicase-assisted replication that is carried out by TWINKLE is not fully understood, but it is clearly of substantial importance, because mutations in TWINKLE are associated with the severe genetic diseases, progressive external ophthalmoplegia, and infantile-onset spinocerebellar ataxia [130].

Two helicases of the Fanconi Anemia gene family, FANCJ and FANCM, are important for fork stability. FANCJ associates with TOPBP1 at sites of replication stress to properly signal the stress response and to limit fork reversal at these sites [131,132]. FANCM is able to migrate replication forks, and is required for fork stability during replication stress [124,133]. Mph1, the yeast homolog of FANCM, can also reverse replication forks, and this activity is regulated in part by the SMC5-6 complex [134,135,136]. The HELQ helicase is required for normal DNA replication, and interacts with the recombination mediators, RAD51, and RAD51 paralogs of the BCDX complex [98]. In HELQ-deficient cells, RAD51 accumulates normally at replication forks after induction of replication stress, but remains bound there, indicating a defect in replication-associated recombination. This inability to resolve stalled replication forks is linked to attrition of germ cells and tumor susceptibility of HELQ-deficient mice.

## 7. Conclusions

Although it is clear that helicases play essential roles in HR, understanding the importance of specific helicases in different recombination processes will require substantial further research. For example, as discussed above, BLM helicase has been reported to function in DNA end resection, displacement of RAD51 from nucleoprotein filaments, disassembly of D-loops, and dissolution of Holliday junctions. These activities represent both pro- and anti-recombinogenic functions, and it is not clear which are most important for maintenance of genomic integrity by BLM. Likewise, many different helicases have been reported to have branch migration activity in eukaryotic cells, and it is not clear to what extent they act redundantly, or whether they are regulated for specific purposes. Solving these questions can be challenging, because it is not always easy to test whether the biochemical activities of helicase enzymes in vitro are also relevant in cells. Future work to clarify the roles of helicases in recombination will give us a better understanding of how HR operates, and potentially open the way to pharmacological intervention to manipulate helicase activities and achieve useful therapeutic outcomes [137].

## Figures and Tables

**Figure 1 genes-11-00498-f001:**
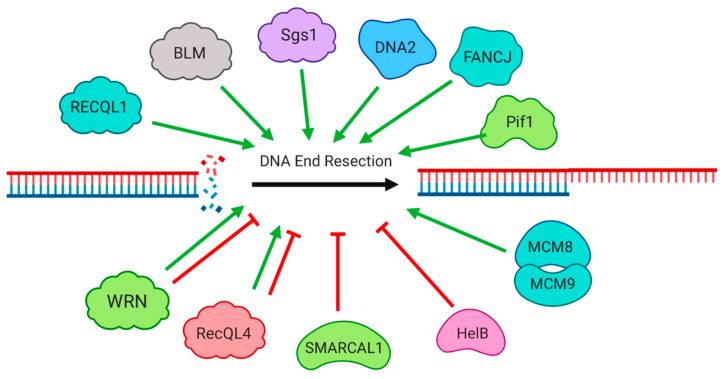
Helicase Proteins involved in the generation of resected DNA ends during recombination. Multiple helicases, such as Bloom Syndrome helicase (BLM) and Sgs1, promote the formation of 3′ single-stranded DNA overhangs necessary for recombination. Other helicases, such as HelB, limit resection, or promote other pathways for repair. For full details, see text.

**Figure 2 genes-11-00498-f002:**
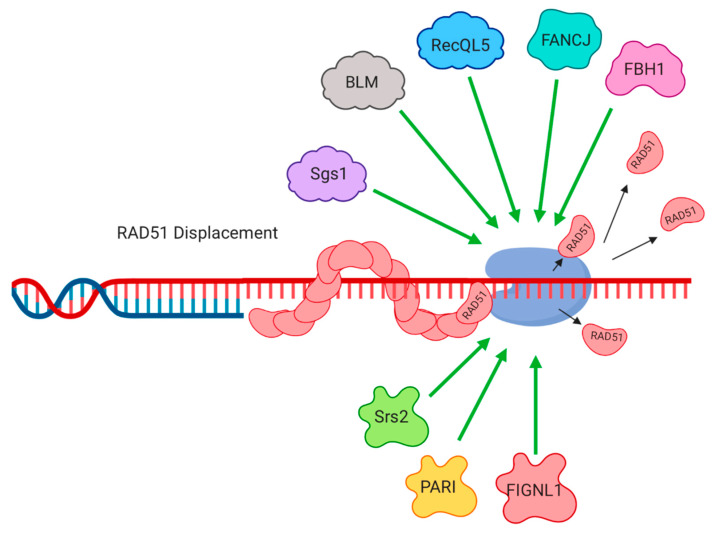
Displacement of RAD51 from resected DNA breaks by helicase proteins. The stability of the RAD51 nucleoprotein filament is regulated by several helicases, which can remove RAD51, thereby reducing the efficiency of recombination.

**Figure 3 genes-11-00498-f003:**
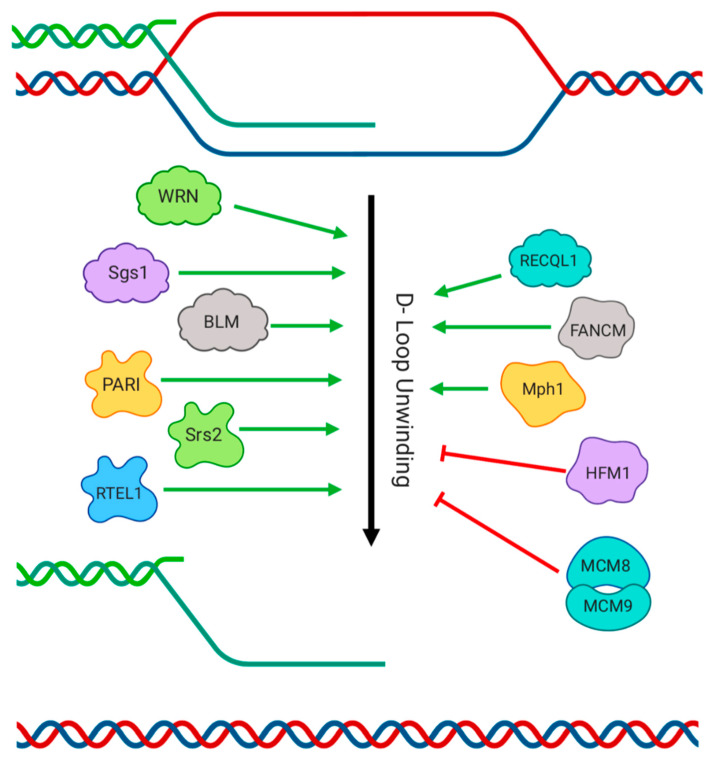
Helicase-mediated unwinding of displacement loop (D-loop) intermediates. Strand invasion of a broken DNA molecule into a homologous duplex creates a D-loop. Template-based repair of sequence at the break site can proceed at the paired 3′ end, using the homologous DNA as a template. This process is inhibited by the action of a number of helicases, such as Srs2 and RTEL1, which exhibit ‘anti-recombinase’ activity by unwinding the D-loop. Other helicases, such as HFM1 and MCM8-9, stabilize the D-loop by supporting DNA polymerase activity, increasing the amount of paired heteroduplex DNA.

**Figure 4 genes-11-00498-f004:**
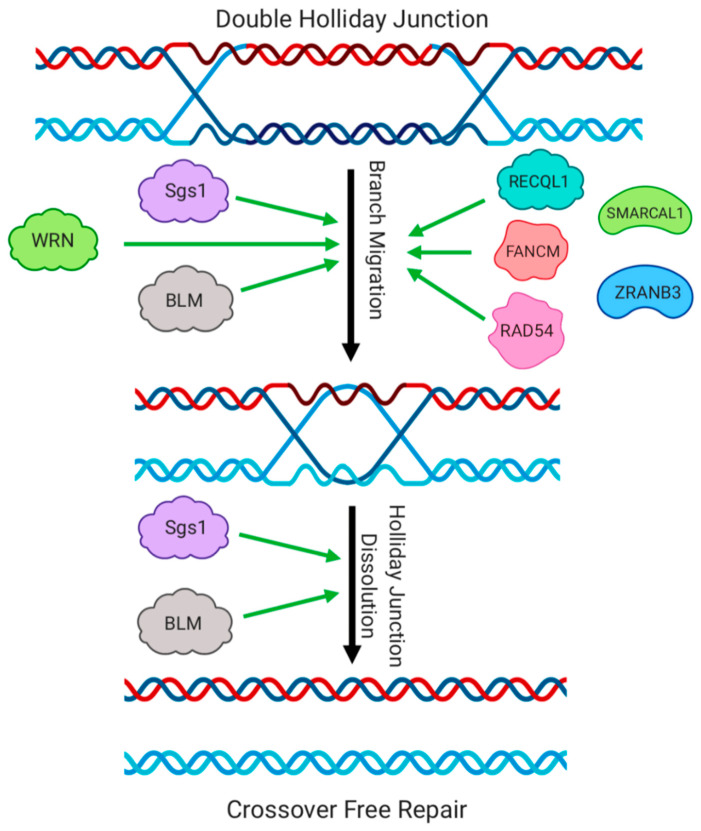
Regulation of Holliday Junction disassembly by helicases. Double-Holliday junctions can be moved into close proximity by the branch migration activity of several helicase molecules. Protein complexes formed by helicases such as BLM and Sgs1 promote dissolution of the hemicatenane intermediate produced by branch migration, leading to non-crossover products of recombination.

**Figure 5 genes-11-00498-f005:**
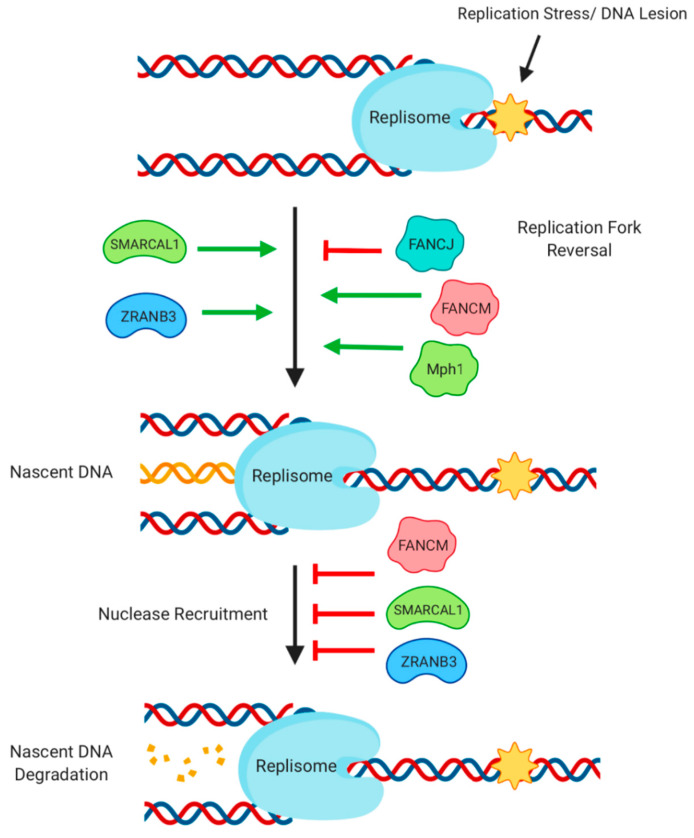
Activity of helicases during DNA replication and transcription. Replication stress caused by a block in replisome progress at a DNA break or lesion can trigger replication fork reversal, which is regulated by a number of helicases. Newly synthesized DNA at the reversed fork is bound by RAD51, protecting it from nucleolytic degradation. Protection of nascent DNA is supported by the presence of several helicases.

**Table 1 genes-11-00498-t001:** Helicases involved in Homologous Recombination.

Family	Gene	Key Function	Species
RecQ	RecQ	Promotes End Resection, D-Loop Disruption, Holliday Junction Migration and Dissolution	*E. coli*
	Sgs1	Promotes End Resection, RAD51 Displacement, D-Loop Disruption, Holliday Junction Migration and Dissolution	*S. cerevisiae*
	BLM	Promotes End Resection, RAD51 Displacement, D-Loop Disruption, Holliday Junction Migration and Dissolution	Mammalian
	WRN	Regulates End Resection, Migrates Holliday Junctions	Mammalian
	RECQL1	Promotes End Resection, D-Loop Disruption, Migrates Holliday Junctions	Mammalian
	RECQL4	Promotes End Resection in S/G2, Suppresses End Resection in G1	Mammalian
	RECQL5	RAD51 Displacement	Mammalian
UvrD	UvrD	RecA Displacement,	*E. coli*
	SLFN11	Replication Fork Signaling	Mammalian
	Srs2	RAD51 Displacement, D-loop Disruption, Holliday Junction Migration	*S. cerevisiae*
	FBH1	RAD51 Displacement and Degradation, Replication Fork Signaling	Mammalian
	PARI	RAD51 Displacement, D-loop Disruption	Mammalian
	HELB	Suppresses End Resection	Mammalian
Fe-S	FANCJ	Promotes End Resection, RAD51 Disruption, Replication Fork Reversal	Mammalian
	RTEL1	D-Loop Disruption	Mammalian
	DNA2	Promotes End Resection.	Mammalian
DEAH Box	FANCM	Replication Fork Reversal, D-Loop Disruption	Mammalian
	FANCJ	Promotes End Resection, RAD51 Disruption, Replication Fork Reversal	Mammalian
	RTEL1	D-Loop Disruption	Mammalian
	Mph1	Replication Fork Reversal, D-Loop Disruption	*S. cerevisiae*
	POLQ	Mediates alt-NHEJ, Displaces RPA	Mammalian
	HELQ	Promotes HR during Replication Stress, Post-Synaptic Recombination Suppression	Mammalian
	HFM1	Meiotic D-Loop Stabilization	Mammalian
SNF2/SWI2-like	RAD54	RAD51 Displacement, Holliday Junction Migration and Dissolution, Promotes D-Loop Formation	Mammalian
	Fun30	Promotes End Resection	*S. cerevisiae* MammalianMammalian
	SMARCAD1	Promotes End Resection	
	SMARCAL1	Regulates End Resection, Replication Fork Reversal, Holliday Junction Migration, Strand Annealing	
	ATRX	Histone H3.3 Replacement, Post-Synaptic Regulation	
MCM	MCM8-9	Promotes End Resection, D-Loop Disruption	Mammalian
AAA ATPase	FIGNL1	RAD51 Displacement	Mammalian
Other	PIF1	Complex Substrate Unwinding	Mammalian/*S. cerevisiae*

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
