# Peer review of "The Regulation of Homologous Recombination by Helicases"

_genes, 2020, doi:10.3390/genes11050498_

Round 1
Reviewer 1 Report
DNA helicases and motor proteins function during multiple steps of homologous recombination (HR), including end resection, regulation of RAD51 nucleoprotein filaments and disassembly of recombination intermediates. Here, the authors provide a comprehensive review of the many DNA helicases that function during HR, and in replication fork remodeling and repair.
Comments:
It might be useful to make the distinction in the Introduction between typical DNA helicases that translocate on ssDNA to unwind dsDNA and motor proteins (SNF2/SWI2 family) that translocate on dsDNA.
Table 1: I suggest using RAD51 in caps consistently in the Table. DNA2 also has an Fe-S cluster. I suggest using SNF2/SWI2 like instead of RAD54 like; RAD54B and Rdh54 should also be mentioned. Fun30/SMARCAD1 is missing from the Table.
End Resection section: Fun30/SMARCAD1 needs to be mentioned for its role in extensive resection. The role of RPA in resection has also been demonstrated in vivo by conditional depletion of RPA70 (Chen et al, 2013). FANCJ has recently been shown to interact with CtIP suggesting a role during the initial step of resection (Nath & Nagaraju, 2020).
Line 115: The term recombination mediator is generally used for proteins that facilitate RAD51 nucleprotein assembly, such as BRCA2 and RAD51 paralogs.
Line 212: The junction formed after strand invasion is not a true Holliday junction.
Reviewer 2 Report
The review by Huselid and Bunting represents nice and up-to-date overview of role of helicases in regulation of homologous recombination (HR). It is divided into logical sections by describing regulation of DNA resection, displacement of ssDNA binding proteins, dissolution of D-loops, role of helicases in the postsynaptic stages of HR and sites of replication and transcription. Since HR is a key pathway required not only for repair of DSBs but also protection of stalled replication forks, understanding its regulation may represent possible way to pharmacologically intervene with this process to prevent or promote genome instability.
Bellow are suggested several comments to help improve this review:
1) The Table 1 should contain also column for type of organism for clarity. ATRX is included in Other family but should by part of RAD54 like family (or rather SWI/SNF).
2) Srs2 activity is not only limited to removing RAD51 from nucleoprotein filament at DSBs as its activity is associated with replication forks where it is actively recruited by interaction with PCNA (Pfander et al 2005, Papouli et al 2005).
3) The stimulation of RAD51 ATPase activity by PARI is observed only at stoichiometric amounts, questioning its biological relevance. In my view PARI does not seem to posses really enzymatic activity displacing RAD51 from ssDNA, but rather regulates HR by limiting the D-loop extension (Burkovicz et al 2016) similarly as for Srs2.
4) The review would benefit if more outstanding questions in the field would be mentioned through the text.
5) Since all of these ezymes are translocases they can dislodge ssDNA from synthetic D-loop substrates and these result has to be interpret with cautious.
6) I would suggest to remove the paragraph describing the transcription-asociated recombination as it rather seems to represent sites of replication transcription collisions and would need to be discuss in this context.
7) The introduction part should contain brief description of HR at replication forks preventing genome instability.
Minor points:
p.1, l.28 should be translocate instead of move along the molecule.
p.4, l. 109 The sentence needs to reformatted to reflect rather their removal from DNA.
p.6, l. 181, should be D-loop.
p.6, l. 179 and 181 and others. Consistency of labelling: RTEL1/RTEL.
p.8, l. 234 C. elegans should be in italic.
p7., l. 219, should be breaks
p.8, l.248 " from DNA" should state dsDNA. The same also p.8, l.261.
p.10. Also HLTF should be mentioned among the replication fork remodellers.
Author Response
Response to Reviewer #2
1) “The Table 1 should contain also column for type of organism for clarity. ATRX is included in Other family but should by part of RAD54 like family (or rather SWI/SNF).”
A column detailing what organism the relevant enzymes are from is now included. ATRX has been moved to the (now-renamed) SWI/SNF2 section.
2) ”Srs2 activity is not only limited to removing RAD51 from nucleoprotein filament at DSBs as its activity is associated with replication forks where it is actively recruited by interaction with PCNA (Pfander et al 2005, Papouli et al 2005).”
We have added discussion of the relevance of Srs2 for replication, and cited these two papers.
3) “The stimulation of RAD51 ATPase activity by PARI is observed only at stoichiometric amounts, questioning its biological relevance. In my view PARI does not seem to posses really enzymatic activity displacing RAD51 from ssDNA, but rather regulates HR by limiting the D-loop extension (Burkovicz et al 2016) similarly as for Srs2.”
This is a great point. I agree that the ability of PARI to displace RAD51 is somewhat uncertain. I have re-written this section to discuss the uncertainties invovled here, and clarify that the mechanism for PARI activity is still an outstanding question.
4) “The review would benefit if more outstanding questions in the field would be mentioned through the text.”
I agree with the reviewer’s suggestion. In general it is beneficial to note active questions in the field, to try to avoid the risk of the article becoming a monotonous list of facts. There are general issues about the molecular mechanisms of how helicases displace RAD51, and about redundancy, which we note in Line 353. The text has now been edited to note the uncertainty about which RECQ helicases mediate resection (Line 86) and about the extent of redundancy between these helicases (222-223). At line 105, we note uncertainty about how MCM8-9 regulates HR. The issue of whether PARI acts to displace RAD51, or at a later stage of recombination, is stated at Line 136 and 200-201. Uncertainty about how FIGNL1 works is noted at Line 165. The question of how the TWINKLE helicase supports recombination in mitochondria is considered further at Line 337, and we now note the serious genetic diseases associated with mutation of TWINKLE. I hope these modifications go some way to addressing te reviewer’s point.
5) “Since all of these ezymes are translocases they can dislodge ssDNA from synthetic D-loop substrates and these result has to be interpret with cautious.”
I am not 100% clear which enzymes the reviewer is referring to here, but I agree that the ability to infer in vivo cellular activities from observations with synthetic substrates is somewhat risky. We have discussed this further in the text.
6) “I would suggest to remove the paragraph describing the transcription-asociated recombination as it rather seems to represent sites of replication transcription collisions and would need to be discuss in this context.”
In response to the reviewer’s suggestion, we have now removed the section about transcription-associated recombination and the accompanying figure (Figure 5B).
7) “The introduction part should contain brief description of HR at replication forks preventing genome instability.”
Consideration of the importance of HR at replication forks is now included in the introduction, along with appropriate references.
“Minor points:
p.1, l.28 should be translocate instead of move along the molecule.
p.4, l. 109 The sentence needs to reformatted to reflect rather their removal from DNA.
p.6, l. 181, should be D-loop.
p.6, l. 179 and 181 and others. Consistency of labelling: RTEL1/RTEL.
p.8, l. 234 C. elegans should be in italic.
p7., l. 219, should be breaks
p.8, l.248 " from DNA" should state dsDNA. The same also p.8, l.261.
p.10. Also HLTF should be mentioned among the replication fork remodellers.”
All of these corrections have been made. Thanks to the reviewer for the close reading of the text.